# Wechsler Scale Intelligence Testing in Males with Dystrophinopathies: A Review and Meta-Analysis

**DOI:** 10.3390/brainsci12111544

**Published:** 2022-11-14

**Authors:** Pien M. M. Weerkamp, Eva M. Mol, Dirk J. J. Sweere, Debby G. M. Schrans, R. Jeroen Vermeulen, Sylvia Klinkenberg, Petra P. M. Hurks, Jos G. M. Hendriksen

**Affiliations:** 1Expert Centre for Neurological and Developmental Learning Disabilities, Kempenhaeghe, Sterkselseweg 65, 5591 VE Heeze, The Netherlands; 2School for Mental Health and Neuroscience, Maastricht University, Universiteitssingel 40, UNS40, 6229 ER Maastricht, The Netherlands; 3Klimmendaal Academy, Klimmendaal Rehabilitation Specialists, Heijenoordseweg 5, 6813 GG Arnhem, The Netherlands; 4Mutsaersstichting, Postweg 88, 5915 HB Venlo, The Netherlands; 5Department of Neurology, Maastricht University Medical Centre, 6229 ER Maastricht, The Netherlands

**Keywords:** dystrophinopathies, Wechsler Intelligence Scales, meta-analysis

## Abstract

Background: Intelligence scores in males with Duchenne Muscular Dystrophy (DMD) and Becker Muscular Dystrophy (BMD) remain a major issue in clinical practice. We performed a literature review and meta-analysis to further delineate the intellectual functioning of dystrophinopathies. Method: Published, peer-reviewed articles assessing intelligence, using Wechsler Scales, of males with DMD or BMD were searched from 1960 to 2022. Meta-analysis with random-effects models was conducted, assessing weighted, mean effect sizes of full-scale IQ (FSIQ) scores relative to normative data (Mean = 100, Standard Deviation = 15). Post hoc we analysed differences between performance and verbal intelligence scores. Results: 43 studies were included, reporting data on 1472 males with dystrophinopathies; with FSIQ scores available for 1234 DMD (k = 32) and 101 BMD (k = 7). DMD males score, on average, one standard deviation below average (FSIQ = 84.76) and significantly lower than BMD (FSIQ = 92.11). Compared to a previous meta-analysis published in 2001, we find, on average, significantly higher FSIQ scores in DMD. Conclusion: Males with Duchenne have, on average, significantly lower FSIQ scores than BMD males and the general population. Clinicians must consider lower intelligence in dystrophinopathies to ensure good clinical practice.

## 1. Introduction

Dystrophinopathies are X-linked recessive progressive neuromuscular disorders caused by mutations in the dystrophin gene. Due to dystrophin deficiency in muscles, patients with dystrophinopathies show progressive muscular weakness, cardiomyopathy, motor delay, and respiratory impairment [1]. The most common and severe type of dystrophinopathy, Duchenne Muscular Dystrophy (DMD), is characterized by a complete disruption of the genetic code in the dystrophin protein and has an estimated birth prevalence of 1 per 3500 to 1 per 9300 males (OMIM: 310,200, ORPHA: 98,896). Boys with DMD usually become wheelchair bound early in the second decade of life [2]. The milder but more variable type of dystrophinopathy, Becker Muscular Dystrophy (BMD), is characterized by a partial disruption of the genetic code in this protein and has a prevalence of 5 per 100,000 males [3]. Motor symptoms in BMD vary from isolated muscle cramps with few effects on ambulation, to being wheelchair bound at the end of the second decade of life [2,3].

Dystrophin can be found in the central nervous system (CNS), muscle cells, and endothelial cells. In the brain, dystrophin is found in regions such as the hippocampus, amygdala, cerebral cortex, prefrontal cortex, and in the Purkinje cells of the cerebellum [4,5,6,7,8]. Its absence is thought to be responsible for the presence of a diversity of neurocognitive symptoms [9] often found in patients with dystrophinopathies [1,5,10,11,12,13,14,15,16,17,18,19]. Previous research has shown an association between the location of the dystrophin gene mutation, affecting specific dystrophin isoforms in the brain (i.e., Dp140 and Dp71), and cognitive impairments. However, the exact phenotype-genotype relationship in both DMD and BMD remains unclear [11,13,20,21,22,23,24]. The study by Cotton, Voudouris, and Greenwood, published in 2001 [25], was the first meta-analysis describing (impaired) intelligence in DMD. Intelligence is the ability to understand and adapt to the environment and to learn from experience [26]. The meta-analysis of Cotton is often used as a key reference in literature on DMD [5,8,12,27,28,29]. It is based on 32 studies published between 1960 and 1999, reporting data on 1224 boys with DMD (mean age = 12.26, SD = 4.06, range 2–27). The data presented are based on scores from eight or more (in some articles, the authors did not explain which instrument(s) is (are) used) different types of intelligence tests (Inc. Stanford-Binet Intelligence Scale, Griffiths Scales, Raven’s Progressive Matrices, Rorsach Developmental Index, Cattel Infant Intelligence Scale, Peabody Picture Vocabulary Test, Wechsler scales, Bayley Scales). Mean full-scale IQ (FSIQ) was 80.20 (SD = 19.3), which is more than one standard deviation below the normal population (M = 100; SD = 15).

In contrast to DMD, less research was conducted on measuring intelligence in BMD boys. A recent review by Ferrero and Rossi (2022) [30] reported on six articles since 1995 on intelligence in BMD. They reported mean FSIQ scores of males with BMD to be (low) average [31,32], where the FSIQ ranged from 82.8 to 95.6. An explanation for this limited number of studies on intelligence in BMD, as opposed to DMD, could be that BMD is characterized by, in general, a milder yet more heterogenous phenotype compared to DMD [3]. 

In research and clinical practice, Wechsler Intelligence Scales are internationally the most frequently used tests to determine intelligence [33], even in DMD males with motor impairments [34]. The Wechsler Intelligence Scales are based on Spearman’s taxonomy of intelligence (1923, 1927), which argued that there are two factors underlying intelligence test performances: (1) a general intelligence factor (*g*-factor) that determines an individual’s “general” abilities level and (2) specific cognitive functions (*s*-factors) that are used to solve specific problems, tasks, and (sub)tests. The current paper aims to review the results of the Wechsler Intelligence Scales administered to males with dystrophinopathies. To our knowledge, this will be the first meta-analysis on Wechsler test scores in BMD. 

The main reason to focus on Wechsler Intelligence Scales alone is to ensure consistency of test scores and therefore minimalizing variation of methods/instruments used. For example, some instruments, e.g., the Raven Standard Progressive Matrices, show only moderate correlation with the Wechsler Intelligence Scales (e.g., the correlation between the Raven Standard Progressive Matrices and Wechsler Intelligence Scales is *r* = 0.55) [35]. Analysing scores from one measure as opposed to comparing scores of different instruments that are, in some cases, not that comparable in what they measure (for instance, they measure different specific cognitive functions), will help uniform data on IQ in DMD and BMD. Therefore, we will re-analyse the data on intelligence test scores—while including studies published until 2022—using only data from studies using the Wechsler Intelligence scales. 

Nonetheless, it should be noted that the Wechsler Intelligence scales have been revised and updated with new subtests and new normative data several times, resulting in a variability of versions since the first (adult) version of this test was published in 1955. There is a separate Wechsler Intelligence scales version for adults (Wechsler Adult Intelligence Scale: WAIS), children (Wechsler Intelligence Scale for Children: WISC), and preschoolers (Wechsler Preschool and Primary Scale of Intelligence: WPPSI) (Wechsler, 2012; Wechsler, 2018; Wechsler, 2020) [35,36,37], covering the whole age range from 2½ years into late adulthood (until 90 years). Moreover, the internal structure of Wechsler Intelligence Scales has also been updated as knowledge of intelligence theory, psychometrics, and diagnostics developed [35]. For example, earlier versions of the Wechsler scales provided FSIQ scores as well as the indices verbal IQ (VIQ) and performance IQ (PIQ). While most recent versions of the Wechsler scales (WAIS-IV, WISC-V, and WPPSI-IV) are based on the theoretical model of Cattell, Horn, Carol (CHC) theory, which is based on Spearman’s two-factor theory. Next to FSIQ (*g*-factor), the WISC-V and WPPSI-IV administer five index scores (*s*-factors): verbal comprehension, visuospatial functioning, fluid reasoning, working memory, and information processing speed, and the WAIS-IV four factors: verbal comprehension, perceptual reasoning, working memory, and processing speed. Additionally, subtests have been removed and introduced in the updated versions. Despite the variability between the structure of the different Wechsler Intelligence scales, correlations between FSIQ scores of different versions (e.g., WISC-V and WAIS-V (*r* = 0.89), WISC-III and WISC-V (*r* = 0.87), WISC-V and WPPSI-III (*r* = 0.78)) are high, indicating that scores on different versions are strongly related within the general population [36]. Nevertheless, in our meta-analyses, we will take the different versions into account as a possible moderator effect.

Despite the abandonment of the PIQ-VIQ split in the newest versions of the Wechsler scales, we decided to do a tentative analysis of PIQ and VIQ in our sample of studies. Cotton et al. (2001) [25] reported, on average, a discrepancy between VIQ (80.40, SD = 18.44) and PIQ (85.40, SD = 16.90) in DMD, which is in accordance with other research and clinical practice showing more language-related problems and dyslexia in DMD. In BMD, there seems to be no significant mean difference in VIQ and PIQ [11,38]. We aim to give an update on the VIQ-PIQ split in DMD and BMD, as this is a clinically relevant split that is reformulated in the newest versions as verbal comprehension and perceptual organization. VIQ and PIQ in dystrophinopathies are still reported in recent studies [23,24,39].

Despite the current sizable literature (on mainly DMD), intelligence scores in dystrophinopathies are still a matter of debate. The main purpose of this meta-analysis is to evaluate the intelligence scores of patients with DMD and BMD, through a systematic review and meta-analysis of published scientific articles over the last decades. To provide better insight into the intelligence scores distribution of the different phenotypes, test scores of, respectively the DMD and BMD population will be compared to normative data of the typically developing population. We will take the mean age of the participants, the publication year of the study, the continent where the study took place, the type of Wechsler Intelligence scale used (WPPSI, WISC, WAIS), and the version of Wechsler Intelligence scale (e.g., WISC-V) into account as possible moderator effects. Results from this review will also be compared to a meta-analysis in 2001 [25] to give an update on the intelligence scores of male patients diagnosed with DMD and BMD. Next to that, we will make a comparison between the intelligence scores of males with DMD and BMD as it is believed that in Becker, FSIQ falls within the normal range and males of DMD are more at risk for lower intelligence.

## 2. Materials and Methods

### 2.1. Protocol and Registration 

A research PROSPERO protocol was submitted (CRD42020194365). 

### 2.2. Search Strategy 

We searched for studies that examined or described standardized intelligence scores of boys and young men with Duchenne Muscular Dystrophy (DMD) and Becker Muscular Dystrophy (BMD) on the Wechsler Intelligence scales, published from 1960 through 2022, and written in the English language. Non-peer-reviewed articles, single case studies, dissertations/thesis, or proceedings of the conference were excluded. To address the risk of bias at the study level: solely Wechsler Intelligence scales were compared and all studies with a biased study sample were excluded (e.g., studies that included only DMD patients with intellectual disabilities or only DMD patients with a certain gene mutation, as those limited samples may not accurately represent the characteristics of the larger population). Studies that did not use Wechsler Intelligence scales to determine intelligence scores or studies that did not publish standardized intelligence scores were excluded. Relevant studies were identified by searching the electronic database PubMed/Medline, Embase, PsychInfo, and Web of Science. Articles were identified by search terms ‘Duchenne’; ‘Muscular Dystrophy, Duchenne’; ‘Becker Muscular Dystrophy’; ‘intelligence’ OR ‘IQ’; ‘intelligence tests’; ‘Wechsler’; ‘WAIS; ‘WPPSI; ‘WISC; ‘TIQ’, ‘FSIQ’. Each search was limited to ‘English’. The first search was set from 1960 to 2020. As there was a delay in our study, we did a second search with the same search strategy from 1960 to March 2022. 

All identified articles were independently reviewed by the first authors (EM, PW) using the Rayyan app (Cambridge, MA, USA) for systematic reviews [40]. Full-text articles were searched and inspected if inclusion or exclusion could not be determined by reading the title or abstract. Cohen’s Kappa statistical analysis was used to measure the inter-rater reliability (≥0.60) of the independent study selection. Disagreements between the two raters after full-text review were resolved by the consensus and use of the last author (JH). PRISMA-P guidelines on meta-analysis [41] were followed.

### 2.3. Data Abstraction

The first author (EM) extracted data from included studies using a standardized form. Data were collected on population (dystrophinophaties, DMD, BMD), sample size (*n*), age (range, mean, SD), location (continent, country), Wechsler Intelligence scale being used (WPPSI, WISC, WAIS, and the different versions). Additionally, results of intelligence testing that were reported as an intelligence quotient (Mean = 100; SD = 15) were searched for. Range, mean, and standard deviation of scores were reported for full-scale (FSIQ), verbal (VIQ), and performance intelligence quotients (PIQ). The first author (PW) checked the extracted data to maximize reliability. To reduce the risk of bias and increase the directness of evidence, in- and exclusion criteria of the individual study samples were assessed. Inclusion criteria that selected a biased sample of the dystrophinopathies were eliminated from our statistical analyses. For example, studies that reported intelligence quotient scores of only a subgroup of the DMD population, such as boys with DMD with comorbid Autism Spectrum Disorder (ASD), were excluded.

### 2.4. Quality Assessment of Studies 

The quality of evidence within the included studies was independently assessed by two authors (PW and DS) with “The Joanna Briggs Institute (JBI) Critical Appraisal Tool for Analytical Cross Sectional Studies [42]. The study design, and analysis method are rated to judge the methodological quality and the extent to which a study has addressed the possibility of bias. This method is recommended for observational, cross-sectional studies [43]. Differences in opinion between the reviewers were discussed until a consensus was reached. 

### 2.5. Statistical Analysis

Descriptive statistics were calculated using IBM SPSS statistics 27 (IBM corp, Armonk, NY, USA). Meta-analyses were performed in RStudio (Posit, PBC, Boston, MA, USA) [44] using the ‘meta’ package [45]. First, for each intelligence quotient outcome measure (FSIQ, PIQ, and VIQ) per subgroup (DMD and BMD), a separate random-effects model meta-analysis was performed, resulting in six separate random effects meta-analyses. In all six models, effect sizes were reported in terms of a standardized mean difference (SMD) using Cohen’s d [46] and once again in terms of the mean difference (MD) to be able to report corresponding intelligence quotient scores. Total study heterogeneity was quantified in terms of the *I*^2^ value and Cochran’s Q tests were performed for all six models to check for significant between-study heterogeneity. Besides 95% confidence intervals (95% CI), also 95% prediction intervals (95% PI) are reported. The 95% prediction interval reflects the range of expected IQ scores in future DMD and BMD patients [47].

Only a few included studies (k = 9) assessed the FSIQ of both healthy controls and males with DMD or BMD. To analyse a possible deviation of the FSIQ scores in dystrophinopathies from the standardization sample, for each included study Cohen’s d was calculated based on the difference between the reported mean FSIQ score and the mean FSIQ score of 100 (SD = 15) as reported for the standardization sample in the manuals of the Wechsler scales [35,36,37]. 

As mentioned above, the newest versions of the Wechsler scales report next to FSIQ, scores on five/four index scores. However, as only two studies made use of a newer version (i.e., WISC-IV) we did not further analyse these index scores of males with dystrophinopathies.

Next, post hoc subgroup analyses were conducted using Q-tests for subgroup differences to analyse possible differences in FSIQ between included DMD and BMD studies and possible differences between reported VIQ and PIQ outcomes within both the DMD and BMD subgroups separately. A post hoc independent samples *t*-test was performed to check whether the reported mean FSIQ scores significantly deviated from the mean FSIQ scores reported by Cotton et al. (2001). Post hoc meta-regression analyses were used to perform moderator analyses for the continent the study took place (i.e., Europe, North America, South America, Asia, and Oceania) to consider possible cross-cultural differences [48,49], the mean age of the included participants, the publication year of the study, the type of Wechsler Intelligence scale used (WPPSI, WISC, WAIS), and the version of Wechsler Intelligence scale (e.g., WISC-V). 

Publication bias was assessed by visual inspection of funnel plots and was statistically assessed using Egger’s tests for funnel plot asymmetry [50]. Lastly, missing values analysis at the study level was computed to evaluate whether missing clinical outcomes (from patients that were included but not fully assessed with the Wechsler Intelligence scales) introduced bias in FSIQ of DMD and BMD. Statistical significance for all analyses was set at *p* < 0.05.

## 3. Results

### 3.1. Study Characteristics 

The initial data search identified 325 articles, the second 42 (Figure 1). Of the studies, left after duplicates were removed, 90 met the eligibility criteria for full-text 333 assessment. The inter-rater Kappa reliability between the two first authors was 0.96. A third author (JH) was involved to solve two disagreements. Forty-seven full-text articles were excluded for diverse reasons (e.g., the study did not use Wechsler Intelligence scales) [12,14,16,18,21,28,29,51,52,53,54,55,56,57,58,59,60,61,62,63,64,65,66,67,68,69,70,71,72,73,74,75,76,77,78,79,80,81,82,83,84,85,86,87,88,89,90] and 43 studies were included in the meta-analysis [10,11,13,19,20,22,23,24,38,39,91,92,93,94,95,96,97,98,99,100,101,102,103,104,105,106,107,108,109,110,111,112,113,114,115,116,117,118,119,120] (Figure 1). Thirty-nine articles comprised data on DMD (84.78%), seven articles related to BMD (15.22%). Three articles reported data on both DMD and BMD [13,39,108] and were counted twice (in both samples). Publication dates ranged from 1972 to 2022. Twenty-three studies originated from Europe (53.50%), five from Asia (11.60%), nine from North America (20.90%), three from South America (7.00%), and three from Oceania (7.00%). 

The 43 studies provided data on 1472 patients with dystrophinopathies, 1354 of them with DMD (91.98%), 118 with BMD (8.20%). Sample sizes differed from 3 to 148. Data on FSIQ were available for 1234 patients with DMD (k = 32) and 101 patients with BMD (k = 7). Data on VIQ were available in 35 studies for DMD (*n* = 1107) and six BMD studies (*n* = 91). Data on PIQ were available in 32 studies for DMD (*n*= 1075) and six BMD studies (*n* = 91). 

Seven studies described a FSIQ of the WPPSI and the WISC (16.30%), three studies of the WPPSI, the WISC, and the WAIS (7.00%), eight studies the WISC and the WAIS (18.60%), twenty-two studies the WISC (51.20%), and three studies the WAIS (7.00%). Different Wechsler versions were used: the WPPSI (WPPSI 70.00%, WPPSI-III 30.00%), the WISC (WISC 32.50%, WISC-R 37.50%, WISC-III 20.00%, WISC and WISC-R combined 2.30%, WISC-III and WISC-IV combined 2.30%, WISC-IV 2.30%), and the WAIS (WAIS 71.40%, WAIS-R 7.10%, WAIS-III 21.40%). 

Notation of age data differed per study, 34 studies provided a mean age, age range, and/or SD; therefore, weighted means are reported. The mean age for the total dystrophinopathy group was 11.65 years (SD = 6.61; Mean range = 5.70–38.80) based on 34 studies, for DMD 10.13 years (k = 27; SD = 2.97; Mean range = 5.70–27.30) and for BMD 27.54 years (k = 6; SD = 10.46; Mean range = 11.00–38.80). 

### 3.2. Quality Assessment of Studies

The overall quality of evidence of the 32 included DMD studies was moderate, and for the 7 BMD studies strong. Details of the evaluation results are presented in Appendix A (Table A1 and Table A2).

### 3.3. Main Effects for FSIQ

Boys and men with DMD (k = 32) had, on average, a significantly lower FSIQ score compared to the means reported for the standardization sample (d = −0.98, Z = −15.26, 95% CI: −1.11, −0.85, *p* < 0.001), this translates into an average FSIQ of 84.76 points for individuals with DMD (95% CI: 82.81, 86.70; 95% PI: 76.34, 93.17) (Table 1). Heterogeneity between DMD studies was significant and is shown in Figure 2 (Q = 69.09, *p* < 0.001, *I*^2^ = 55.10%).

Boys and men with BMD (k = 7) also had, on average, a significantly lower FSIQ compared to the means reported for the standardization sample (d = −0.52, Z = −2.54, 95%CI: −0.92, −0.12, p = 0.01), this translates into an average FSIQ of 92.11 points (95%CI: 86.71, 97.50; 95%PI: 79.63, 104.59) (Table 1). Heterogeneity between BMD studies was not significant and is shown in Figure 3 (Q = 10.08, *p =* 0.12, *I*^2^ = 40.50%) (Figure 3).

Our results for the random effect model showed that the FSIQ scores of boys and men with BMD were, on average, significantly higher compared to boys and men with DMD (*Q* = 6.31, on 1 degrees of freedom, *p =* 0.01).

### 3.4. Tentative Post Hoc Analysis for VIQ and PIQ

The VIQ in DMD (*k* = 35) was, on average, significantly lower compared to the means reported for the standardization sample(*d* = −1.04, *Z* =−18.33, 95% CI: −1.15, −0.93; 95% PI: −1.42, −0.66), this translates into an average VIQ of 84.21 points (95% CI: 82.63, 85.78; 95% PI: 78.61, 89.81), see Table 1. Heterogeneity among DMD studies was significant (*Q* = 51.27, *p =* 0.03, *I*^2^ = 33.70%).

The average VIQ in BMD (*k* = 6) was also, on average, significantly lower compared to normative data (*d* = −0.56, *Z* = −1.99, 95% CI: −1.12, −0.01; 95% PI: −2.21, 1.08). This translates into an average VIQ of 90.81 (95% CI: 82.63, 98.99; 95% PI: 66.75, 114.87) (Table 1). Heterogeneity among studies was significant *(Q* = 13.34, *p=* 0.02, *I*^2^ = 62.50%).

PIQ scores in DMD (*k* = 32) were, on average, significantly lower compared to the normal population (*d* = −0.74, *Z* = −11.54, 95% CI: −0.86, −0.61; 95% PI: −1.24, −0.23). This translates into an average PIQ of 88.82. (95% CI: 82.63, 98.99; 95% PI: 66.75, 114.87), see Table 1. Heterogeneity among studies was significant (*Q* = 58.88, *p <* 0.01, *I*^2^ = 47.30%). PIQ scores in BMD (*k* = 6) were, on average, significantly lower compared to the normal population (*d* = −0.74, *Z* = −2.40, 95% CI: −1.35, −0.14; 95% PI: −2.60, 1.12). This translates into an average PIQ of 89.18 (95% CI: 81.33, 97.03; 95% PI: 65.29, 113.07) (Table 1). Heterogeneity among studies was significant (*Q* = 14.02, *p =* 0.02, *I*^2^ = 64.30%).

### 3.5. Moderator Analyses

For DMD, significant moderator effects were found in the FSIQ data regarding the continent where the study took place (R^2^ = 60.27). Studies taking place in South America showed significantly lower average FSIQ scores compared to studies from Europe (Z = 4.01, SE = 0.20 *p* < 0.001), Asia (Z = 2.70, SE = 0.24, *p* < 0.01), North America (Z = 4.92, SE = 0.17, *p* < 0.001), and Oceania (Z = 3.08, SE = 0.20, *p* = 0.002). FSIQ scores from the other continents did not significantly differ between one another, *p* > 0.05. No significant moderator effect was found for the continent where the study took place when studying the FSIQ data of the BMD group (*p* > 0.05). In de BMD group, a significant moderator effect was found on FSIQ for the WISC version used (Z = 2.12, SE = 0.47, *p* = 0.03): FSIQ scores on WISC-III are, on average, significantly higher than those scores on WISC-R, in BMD. No significant moderator effect was found on FSIQ for the other used versions (i.e., the type of the Wechsler Intelligence scale used (WPPSI, WISC, WAIS) and the version of the Wechsler Intelligence scale (e.g., WISC-III and WISC-IV)) in the BMD group (*p* > 0.05). No type and version effects were found in the DMD sample (*p* > 0.05).

The other moderator effects being analysed were: the mean age of the participants and the publication year of the study. Both were also not significant in both groups (*p* > 0.05).

### 3.6. Publication Bias

Egger’s regression did not indicate a publication bias concerning FSIQ scores in DMD (*t* = 1.28, *p* = 0.21). As *p* > 0.05, an additional trim-and-fill analysis was not conducted. See Figure 4 for the funnel plot.

Following recommendations by Sterne et al. (2011) [41], by default, a test for funnel plot asymmetry is only conducted if the number of studies is ten or more (k·min = 10). Therefore, the funnel plot and Egger’s test of FSIQ in BMD should be considered with caution (k = 7). Our tentative results indicate that there does not seem to be a publication bias for FSIQ in BMD (*t* = 0.28, *p* = 0.79) (Figure 5).

### 3.7. Missing Values Analysis

No significant missing clinical values were indicated for FSIQ scores in DMD (Z = 0.93, Standard Error = 0.01, *p* = 0.034) nor in BMD (Z = 70.00, Standard Error = 0.53, *p* = 0.48). 

### 3.8. Comparison with Data from Cotton et al. (2001)

The FSIQ in our DMD sample (Mean = 84.67, SD = 34.86) was significantly higher than reported by Cotton (M = 80.20, SD = 19.3), (*t* = −3.90, *p* < 0.001). 

## 4. Discussion

The aim of the current study was to conduct a meta-analysis on the Wechsler Intelligence scale scores of boys and men diagnosed with DMD and BMD. As there is a growing scientific and clinical interest in brain-related comorbidities of DMD and BMD [7], we felt the need to re-analyse available data as published in 2001 (Cotton et al. (ref) who analysed data of 32 studies using more than eight different intelligence tests). Our data describe 42 studies using Wechsler Intelligence scales. We found that males with DMD have a mean FSIQ score of 84.76 (95% CI: 82.81, 86.70; 95% PI: 76.34, 93.17), which confirms the Cotton review that FSIQ in DMD is one standard deviation below the average IQ of the general population and is to be classified as a low average intelligence [31,32]. In comparison with the Cotton review, our data show that the mean intelligence of boys and males with DMD is underestimated with 4.56 FSIQ points. The clinical significance of this statistical significant difference is not yet clear, but is a difference that needs to be taken in consideration when describing new data on intelligence scores in boys and men with dystrophinopathies. Our results predict that there is a 95% chance that a boy or man with DMD will have a FSIQ between 76.34 and 93.17, i.e., a below average to average score. We argue that the current, higher estimate is more robust because of our decision to use Wechsler Intelligence scales only and excluding studies with a selection bias. Furthermore, newer meta-analysis techniques enable us to examine variations in intelligence scores of patients with dystrophinopathy by possible confounding moderator variables such as: age of the participants, publication year of the study, type of Wechsler scale used (WPPSI, WISC, WAIS), version of Wechsler scale (e.g., WISC-V), and the location/continent where the article was published. Except for continent, none of these moderator variables interfered significantly with intelligence scores in our DMD group. In South-America, intelligence scores of boys and young men with DMD seem to be significantly lower than in other continents. However, it should be noted that only two South-American studies were included, both from Brazil, and that one of the two studies is identified as a significant outlier, the study of Rapaport et al. (1991) [93] reports a mean intelligence score of 71.76 (*n* = 148). However, we did not observe any differences in study methodology to explain this. We found that European studies were overrepresented in this review, which seems to be an overall trend in literature on DMD [121]. 

This is the first study analysing the intelligence scores of in a total of 101 boys and men with BMD in 7 studies using Wechsler Intelligence scales. Results indicate that BMD boys and men have a mean FSIQ of 92.11, an average intelligence score. The reliability interval of the intelligence scores of boys and men with BMD predicts that there is 95% chance a male with BMD will have a FSIQ on the Wechsler Intelligence scales between 79.63 and 104.59. The mean FSIQ scores of the BMD population were statistically significantly higher than those of boys and men with DMD. Where males with BMD score on average 7.35 FSIQ points more. However, it should be noted that the mean age of males with BMD was higher (27.54 years) than of our DMD population (10.13 years). Previous research has shown intelligence scores to stabilize from the age of 11–12 years [122,123,124], which is a little older than our DMD sample. As age was not a moderator in our meta-regression; therefore, we believe our FSIQ difference between DMD and BMD is representative.

Earlier research reported a statistically significant discrepancy between verbal IQ and performance IQ scores in boys and young men with DMD [25,99,110]. Our tentative analysis also suggests a statistically significant discrepancy between verbal IQ (84.21) and performance IQ (88.82) scores in boys and young men with DMD. The gap reported is similar to that reported by Cotton et al. (verbal IQ: 80.40 and performance IQ: 85.40). Several studies have supported the hypothesis that the lack of dystrophin in the cerebellum and hippocampus may play a role in language, (working) memory, and reading deficits [9,17,54,78,91,125]. In BMD, no discrepancy was found between verbal and performance IQ (mean VIQ: 90.81 and mean PIQ: 89.18). In the more recent theoretical models on intelligence (CHC model; WPPSI-IV, WISC-V and WAIS-V) the VIQ/PIQ structure is abandoned and exchanged for more specific intelligence index scores such as working memory. This an important domain to study within the DMD population since (verbal) working memory problems have been previously reported [51,80,125]. 

One important moderator factor, that has not been considered in both our review and the Cotton review, is the degree of (fine) motor impairment in boys and young men as some of the subtests of the Wechsler Intelligence scales require motor function and speed of (fine) motor function. Patients with dystrophinopathies may be hindered by progressive muscular weakness and motor delay. Different studies show that patients with impaired motor function are at a disadvantage on some of the subtests of the Wechsler Intelligence scales in which motor skills are required, leading to an underestimation of their intelligence scores [126,127]. Therefore, Pioversana et al. (2019) [128] developed an alternative motor-free Wechsler Intelligence scale measuring method to the ‘traditional’ test administration, calculating the FSIQ using only subtests from the WISC-V that do not have motor involvement: vocabulary and similarities (verbal comprehension index), visual puzzles and figure weights (perceptual reasoning index), digit span and letter number sequencing (working memory). This motor-free short-form of the WISC-V has not been used until now in the dystrophinopathy population, but it has been proven applicable for adolescents with Cerebral Palsy [129]. In future research, we recommend further exploration of the use on the WISC-V motor-free short-form in boys with DMD and BMD.

An important limitation of our work is the variability in versions of Wechsler scales, i.e., some children were assessed with WISC-R and other children with the WISC-V. Therefore, there may be some differences in intelligence scores that are subsequent to the varying versions that are based on different subtests. Our tentative analysis suggests that there are no confounding effects from the variability in versions, except in the BMD sample, the WISC-III (k = 2) differed from the WISC-R (k = 2, with WISC-R showing lower FSIQ scores). However, all other versions (e.g., WAIS or WISC and WISC-III or WISC-IV) did not differ in both samples. Therefore, we believe our overall intelligence scores are representative for our population. This analyses of moderator effects learns us that there were no significant moderator effect for the different versions of the Wechsler scales being used until now (paragraph 3.5). Therefore, it may be expected that the recent conversions into the WISC-V and WPPI-IV will not result in different intelligence scores for the dystrophinopathy populations.

The overall methodological quality of the included studies is for the DMD studies moderate and for the BMD studies strong, Although some included studies did not have a strong quality of evidence, we included all of the studies in our analysis. Considering that multiple studies received a weak or moderate rating because the authors did not identify or deal with confounding factors. However, intelligence scores were most of the time reported as a descriptive and not part of an intervention or manipulation in the reviewed studies. Nonetheless, it should be noted that some available studies did have limitations that could affect our results. In some cases, small sample sizes affect the precision of the estimated estimates and power of the included study. Secondly, in multiple studies information is missing. It was, often unclear in the reviewed studies whether the intelligence testing was carried out by a trained psychologist or not, which may lead to a ‘moderate’ score. Consequently, the study quality might have been under- of overestimated. Especially within the dystrophinopathy population, it should be taken in account by whom intelligence was assessed as the testing of men with dystrophinopathy may require more than average testing skills due to the complexity of their functioning caused by the brain related comorbidities [7].

## 5. Conclusions

In conclusion, our systematic review and meta-analysis demonstrated that males with DMD may manifest clinically and statistically meaningful reductions in overall intelligence compared to normative data. The mean FSIQ score found in DMD is 4.46 points higher in comparison to the review of the Cotton. On the one hand, our data confirm the findings of the Cotton review; on the other hand, our data further refine and elaborate on intelligence in DMD and BMD. The gap between performance IQ and verbal IQ in boys and men with DMD was not found in the seven studies published on BMD. Males with BMD have average intelligence scores. We recommend to routinely offer intelligence and cognitive functioning screening to patients with DMD and BMD and that the Wechsler Intelligence scales provide a solid instrument to measure intelligence in individuals with dystrophinopathies. These data may contribute to help patients achieve optimal participation in society (e.g., education, work, daily functioning).

## Figures and Tables

**Figure 1 brainsci-12-01544-f001:**
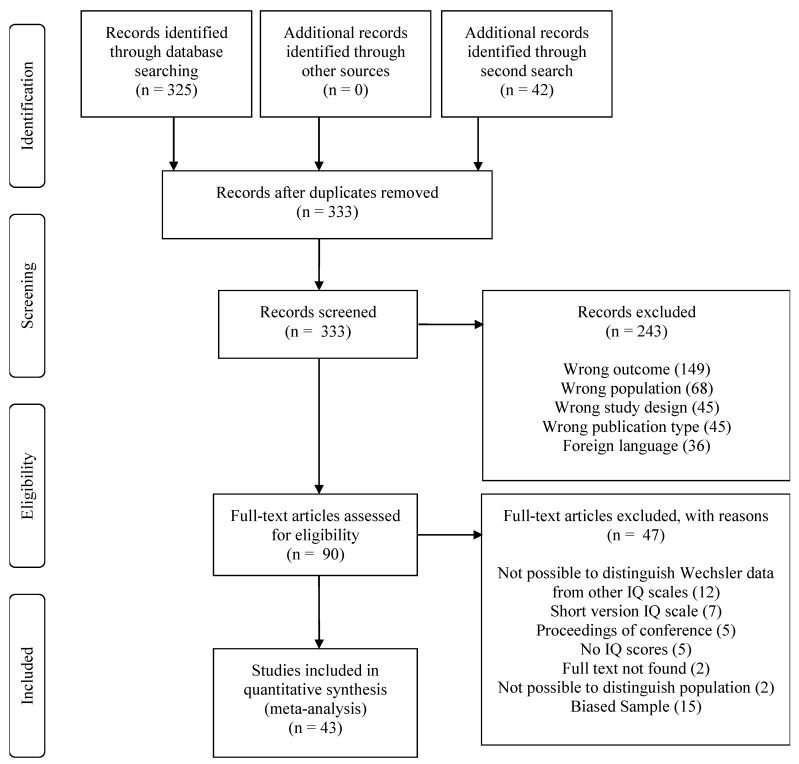
Flowchart of the selected articles.

**Figure 2 brainsci-12-01544-f002:**
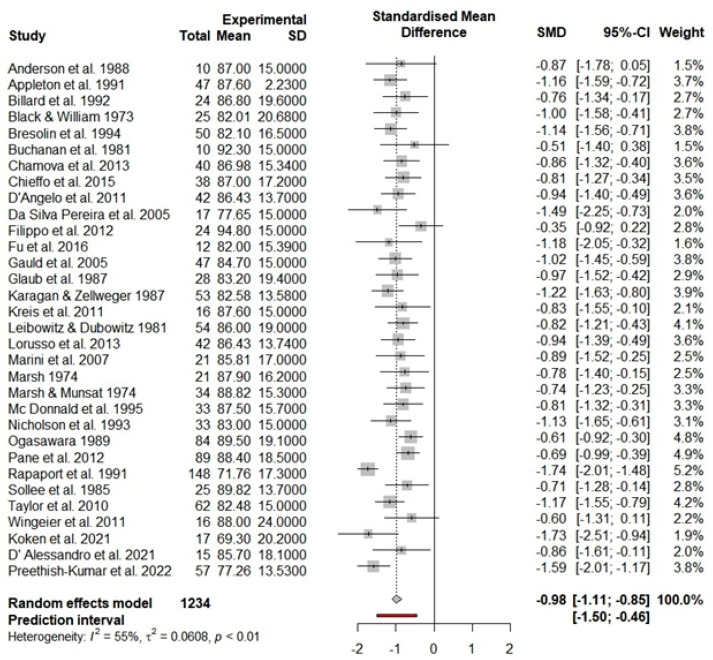
Forest plot of full-scale intelligence scores in the DMD population. Abbreviations used: CI = confidence interval; *p* = significance level; SMD = standard mean difference; SD = standard deviation [10,13,24,26,41,96,97,98,99,101,102,103,104,107,108,109,110,113,114,115,117,118,121,122,123,124,125,126,127,128,129].

**Figure 3 brainsci-12-01544-f003:**
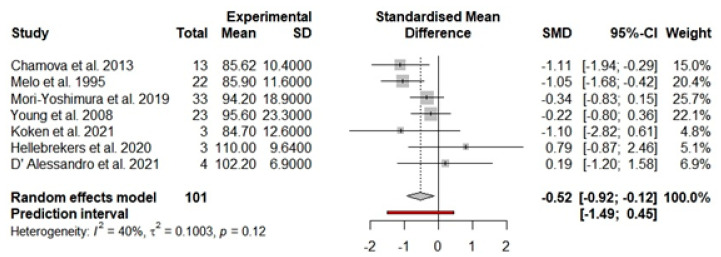
Forest plot of full-scale intelligence scores in the BMD population. Abbreviations used: CI = confidence interval; *p* = significance level; SMD = standard mean difference; SD = standard deviation [[11],[13],[41],[42],[105],[115],[120],].

**Figure 4 brainsci-12-01544-f004:**
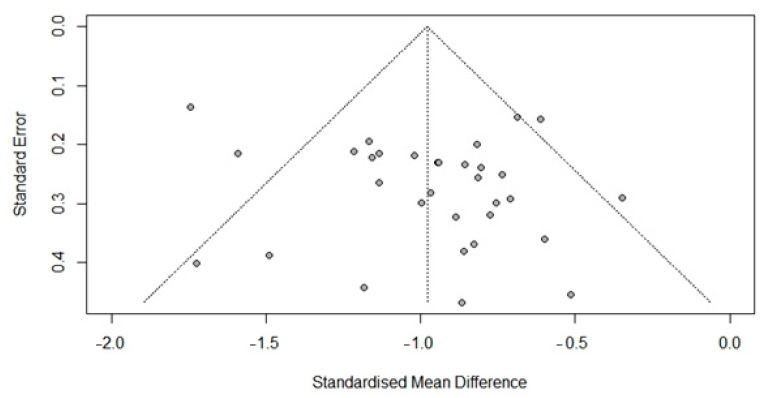
Funnel plot of full-scale intelligence scores in the DMD population.

**Figure 5 brainsci-12-01544-f005:**
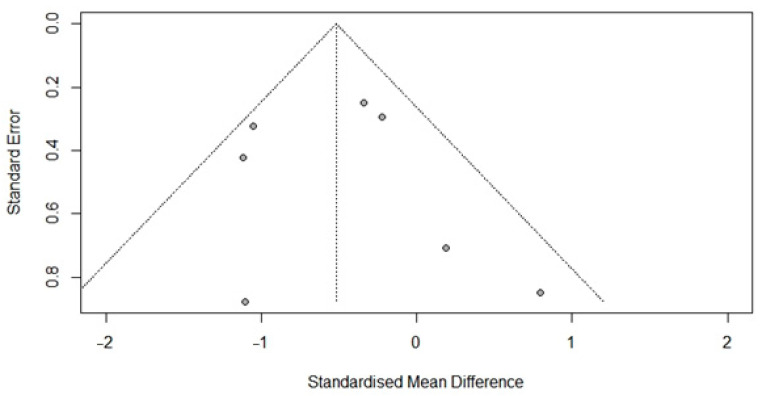
Funnel plot of full-scale intelligence scores in the BMD population.

**Table 1 brainsci-12-01544-t001:** Weighted mean effect size meta-analysis.

				95% CI			Homogeneity	
	*k*	*n*	*M*	*LL*	*UL*	*p*	*Q*	*I* ^2^
DMD FSIQ	32	1234	84.76	82.81	86.70	<0.001	69.09	55.10
BMD FSIQ	7	101	92.11	86.71	97.50	0.12	10.08	40.50
DMD VIQ	35	1107	84.21	83.44	85.95	0.03	51.27	33.70
BMD VIQ	6	91	90.81	82.45	99.70	0.02	13.34	62.50
DMD PIQ	32	1075	88.82	87.55	91.05	<0.01	58.88	47.30
BMD PIQ	6	91	89.18	77.93	95.83	0.02	14.02	64.30
Cotton * FSIQ	32	1145	80.20					

Note: * refers to the review of Cotton et al. (2001). Abbreviations used: BMD = Becker muscular disorder; DMD = Duchenne muscular disorder; FSIQ = full-scale intelligence quotient; *LL* = lower limit; *M* = mean; *p* = significance level; PIQ = performal intelligence quotient; *Q* = heterogeneity statistic; UL = upper limit; VIQ = verbal intelligence quotient.

## Data Availability

Data supporting the findings of this study are available from the corresponding author P.M.M.W. on request.

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
