# Peer review of "Wechsler Scale Intelligence Testing in Males with Dystrophinopathies: A Review and Meta-Analysis"

_brainsci, 2022, doi:10.3390/brainsci12111544_

Round 1
Reviewer 1 Report
The manuscript entitled “Meta-analysis of data on Wechsler scale intelligence testing in 2
males with dystrophinopathies: a review and meta-analysis” sought to elucidate the cognitive profile of males with dystrophinopathies. This would be a useful manuscript for both clinicians and researchers. I like the focus on scores from one measure as opposed to looking at scores across different tests, which, in some cases, are not all that comparable. I congratulate the authors on an interesting manuscript that seems to have largely appropriate methodology and analyses. The manuscript could be improved by clarifying or considering some of the following areas:
· In the introduction it would be interesting for the authors to provide more justification for why we are interested in IQ in this population – they mention that there are neurocognitive symptoms but don’t go much beyond this – adding a couple of sentences saying why we expect cognitive symptoms in this sample and why it is important to look at this would help motivate the study.
· It would also be interesting in the introduction if the authors mention why there are so few studies on IQ in BMD – perhaps there are fewer cognitive symptoms in this sample or because of the heterogeneity of the phenotype?
· The wording on line 51 is confusing “the relation between (impaired) intelligence in Duchenne.” The relationship between intelligence in Duchenne and what?
· One line 64 I would suggest an alternate wording – “the borderline range of average to just above” does not seem to well capture the scores the authors present
· On line 73, perhaps specify the focus on Wechsler FSIQ scores – it’s implied from what comes before but saying it specifically would be clearer
· The sentence on lines 80-82 could be reworked for clarity
· The authors use an abbreviation for the WAIS on line 86 but it does not seem to be written out fully prior – to remain consistent with the other tests you mention, it would be best to write it as Wechsler Adult Intelligence Scale (WAIS)
· On line 92 it is redundant to say “full FSIQ”
· On line 140 the authors say that you excluded studies that contained a limited sample (e.g., those with DMD and intellectual disability). It would be nice to have some more justification for this – some could argue for inclusion of these individuals since they represent part of the spectrum of scores you are interested in
· On line 196 the authors say that they ran an analysis to see whether their results deviated from Cotton et al. I question whether this is meaningful given the age of that study and the fact that scores from different tests were reported. The authors could describe their results in relation to Cotton without this statistical analysis and it would be just as meaningful – the fact that the difference is statistically significant doesn’t really mean that it is clinically significant or meaningful given the above
· On line 199 the authors mention that they used ‘continent’ as a moderator. What is the justification for doing this? Unless there is a theoretical justification, I would not include it as a moderator. The authors end up with a finding that is not really meaningful given that it seems that only 1 outlying study seems to have driven the finding – there aren’t enough studies from the different regions and there does not appear to be any reason to include this as a moderator.
· The age of the samples between DMD and BMD seem quite different – in DMD the sample is primarily children/teens whereas in BMD the sample is primarily adults – although age was not found to be a moderator – it would be good to address this difference in age in the discussion
· In the discussion it would be interesting if the authors could provide more explanation for why there is a discrepancy in VIQ/PIQ in DMD – can they tie this to the neurocognitive symptoms/brain differences/phenotype in this sample?
· The authors mention as a limitation that they were unable to replicate the meta-analysis done by Cotton et al. – replicating this does not seem important given that they are using data from different tests (a more limited sample of tests). Rather than state this as a limitation, the authors could simply mention that their study was not an attempt to replicate Cotton et al since they used different test scores. Their study is instead a separate meta-analysis that updates and refines the Cotton study.
Reviewer 2 Report
Introduction
I think the introduction is adequate, focuses on the research topic, and is easy to follow.
As for the references used in the introduction, they are adequate and current, so I have no additional comments on this section.
Materials and Methods
Protocol and registration
It is a strong point that the manuscript has been submitted and approved in PROSPERO.
Search Strategy
I propose adding "google scholar" to the databases since I think it could provide some additional interesting manuscripts to include in the review.
Data abstraction
I believe that the information included in this section is adequate and sufficient, I do not propose any changes here.
I think it is necessary to include the risk of bias and the quality of evidence (e.g., GRADE).
Statistical analysis
I believe this section is well-written and contains the necessary information
Results
In general, I believe that the results are correct and include the most relevant information, although, as I said before, I would include the risk of bias and the quality of the evidence through a tool that has been previously validated for this purpose.
I think it might be interesting to perform a meta-regression to see by which factors the results of the meta-analysis could be explained, what do you think about this?
Discussion
Overall I think it is correct but would benefit from the introduction of a discussion of the data discussion on parity risk and quality of evidence, please include it in the discussion.
Conclusion
I believe that the conclusion is adequate, although it is not usual to include a quote in the conclusion, please remove it.
References
In references 91, 108,73,83 and 51 much of the text are capitalized, please correct this.
Round 2
Reviewer 2 Report
The authors have responded to all my questions and comments, in my opinion the manuscript can be accepted.